# Cytological Evaluations of Advanced Generations of Interspecific Hybrids between *Allium cepa* and *Allium fistulosum* Showing Resistance to *Stemphylium vesicarium*

**DOI:** 10.3390/genes10030195

**Published:** 2019-03-04

**Authors:** Natalia Kudryavtseva, Michael J. Havey, Lowell Black, Peter Hanson, Pavel Sokolov, Sergey Odintsov, Mikhail Divashuk, Ludmila Khrustaleva

**Affiliations:** 1Center of Molecular Biotechnology, Russian State Agrarian University-Moscow Timiryazev Agricultural Academy (RGAU-MTAA), 49, Timiryazevskaya Str., 127550 Moscow, Russia; natalia.kudryavtseva92@gmail.com (N.K.); pav2395147@yandex.ru (P.S.); sodinc@yandex.ru (S.O.); divashuk@gmail.com (M.D.); 2USDA-ARS and Department of Horticulture, University of Wisconsin, Madison, WI 53706, USA; michael.havey@ars.usda.gov; 3Seminis Vegetable Seeds, DeForest, WI 53532, USA; llblacktw@netscape.net; 4The World Vegetable Center, P.O. Box 42, Shanhua Tainan 74199, Taiwan; peter.hanson@worldveg.org

**Keywords:** Genomic in situ hybridization (GISH), recombination, polyploidization, cytoplasm, *Allium cepa*, *Allium fistulosum*

## Abstract

Interspecific crossing is a promising approach for introgression of valuable traits to develop cultivars with improved characteristics. *Allium fistulosum* L. possesses numerous pest resistances that are lacking in the bulb onion (*Allium cepa* L.), including resistance to *Stemphylium* leaf blight (SLB). Advanced generations were produced by selfing and backcrossing to bulb onions of interspecific hybrids between *A. cepa* and *A. fistulosum* that showed resistance to SLB. Molecular classification of the cytoplasm established that all generations possessed normal (N) male−fertile cytoplasm of bulb onions. Genomic in situ hybridization (GISH) was used to study the chromosomal composition of the advanced generations and showed that most plants were allotetraploids possessing the complete diploid sets of both parental species. Because artificial doubling of chromosomes of the interspecific hybrids was not used, spontaneous polyploidization likely resulted from restitution gametes or somatic doubling. Recombinant chromosomes between *A. cepa* and *A. fistulosum* were identified, revealing that introgression of disease resistances to bulb onion should be possible.

## 1. Introduction

*Stemphylium* leaf blight (SLB), caused by *Stemphylium vesicarium* (Wallr.) Simmons, is a serious fungal disease of bulb onion (*Allium cepa* L.) occurring worldwide, which may result in 100% losses of the bulb crop [1]. SLB has also been reported in garlic [2], asparagus [3], sunflowers [4], pear [5], radish [6], and tomato [7]. Research has been undertaken to identify sources of resistance to SLB in *Allium* species and determine its genetic basis [8]. *Allium fistulosum* L. is a source of desirable traits for improvement of the bulb onion [9] and shows resistance to *Stemphylium vesicarium* which may be conditioned by a single dominant gene [8]. In 2001, hybrids between *A. cepa* and *A. fistulosum* were produced to initiate introgression of resistance to SLB into the bulb onion [8]. Although the first interspecific hybrids between *A. fistulosum* and *A. cepa* were obtained in 1935 [9], hybrids are highly sterile, and numerous attempts to transfer beneficial traits from *A. fistulosum* into the bulb onion have not been successful. The first fertile advanced backcross plants between *A. cepa* and *A. fistulosum* were obtained by Hou and Peffley [10]; however, no commercial cultivar of the bulb onion has ever been developed with a desirable trait from *A. fistulosum*.

Genomic in situ hybridization (GISH) is a reliable technique for the identification of parental genomes at the chromosomal level and for the monitoring of alien gene introgression via interspecific hybrids [11,12,13]. However, the technique is often not used in breeding because of its labor intensity and low-throughput. Molecular markers are an effective method for tracking of valuable traits allowing acceleration of the breeding process. However, molecular markers provide information only across chromosomal regions in linkage disequilibrium with the target trait. A huge number of molecular markers are required to analyze chromosomal introgression from one species to another. GISH provides information at the chromosome level and does not require any knowledge about genome sequences and can confirm copy numbers of homologous chromosomes in polyploid hybrids, which is difficult using molecular markers.

In this paper we present the results of a cytological study of advanced generations derived from hybrids between *A. cepa* × *A. fistulosum* that show resistance to *Stemphylium vesicarium*.

## 2. Materials and Methods

### 2.1. Plant Material

Seeds of parental species (*A. cepa* and *A. fistulosum*) and advanced generations derived from interspecific hybrids [8] were provided by the Asian Vegetable Research and Development Center in Taiwan (Table 1). A landrace (‘Tuwel’) of *Allium fistulosum* from Indonesia (accession AF468) was used as a male parent and *A. cepa* cultivar ‘Rouge de Tana’ (TA207) as the female parent to produce the interspecific hybrid, which was self-pollinated to the S_5_ generation (AVON1275). Backcross generations were produced using AVON1275 and the open-pollinated onion cultivar ‘Arka Niketan’ (AC464), developed by the Indian Institute of Horticultural Research. After backcrossing to Arka Niketan, embryos were rescued [14] and resulting plants were evaluated for *Stemphylium* leaf blight (SLB) resistance after artificial inoculation (described below). Backcross progenies from the same family that showed resistance to SLB were intercrossed. Plants were grown in pots in greenhouses at RGAU-MTAA (Russia) or the University of Wisconsin (USA).

### 2.2. Chromosome Preparations

Mitotic chromosomes were prepared from young root meristems using the squash method according to Khrustaleva and Kik [15] with slight modifications. Young root tips were pretreated overnight with an aqueous saturated solution of 1-bromnaphtalene at 4 °C, fixed in 3:1 (*v/v*) ethanol-acetic acid for at least 1 h at room temperature and then stored at −20 °C, rinsed four times in distilled water, and finally incubated in 10 mM citrate buffer (pH 4.8) containing 0.1% (*w/v*) Pectolyase Y-23 (Kikkoman, Tokyo, Japan), 0.1% (*w/v*) Cellulase Onozuka R-10 (Yakult Co. Ltd., Tokyo, Japan) and 0.1% (*w/v*) Cytohelicase (Sigma-Aldrich, France) for 50 min at 37 °C. The macerated root tips were spread by dissecting and squashing in a drop of 45% acetic acid.

### 2.3. Analysis of Fertility

Pollen morphology was studied using an acetocarmine staining method [16] from flowering plants from AVON 1275, AVON 1290, AVON 1291, AVON 1501, and AVON 1502. The other hybrid lines did not flower under our greenhouse conditions. We analyzed five flowers per accession when flowers were at peak anther maturity. Two anthers from each flower were squashed in 1% acetocarmine on separate microscope slides and analyzed using phase−contrast microscopy with ocular ×12 and objective ×10 magnification.

### 2.4. Genomic DNA Isolation and Probes Preparation

Genomic DNA was isolated from young leaves of the parental species (*A. cepa* TA207and AC464, and *A. fistulosum* AF468) according to the protocol of Rogers and Bendich [17]. Genomic DNA of *A. fistulosum* was sonicated to 1–3 kb fragments and used for the labeled probe preparation. DNA from *A. cepa* was sonicated to 200–400 bp fragments and used as the blocking DNA. Probe DNA was labeled with digoxigenin (DIG)-11-dUTP by nick-translation (Roche, Diagnostics Gmbh, Mannheim, Germany).

### 2.5. GISH

In situ hybridization, immunological detection, and counterstaining procedures were the same as previously described by Khrustaleva and Kik [15]. The hybridization mixture contained: 50% (*v/v*) formamide, 10% (*w/v*) dextran sulfate, 2×SSC, 0.25% (*w/v*) sodium dodecyl sulfate (SDS), 50 ng/µL of labeled probe (DIG)-11- dUTP DNA (A. fistulosum), and 1500 ng/µL of blocking DNA (*A. cepa*). In the hybridization mixture, we used a ratio of 1:30 of probe and block DNA, and washes at 78% stringency were applied. GISH analyses were performed on 3 to 5 plants from each accession.

### 2.6. Microscopy, Image Analysis, and Karyotyping

GISH preparations were visualized using a fluorescent microscope, Zeiss Axio Imager (Carl Zeiss MicroImaging, Jena, Germany), conjugated with black−white sensitive digital camera Axiocam, and with Axio Vision software v. 4.6.3. The final optimization of images was performed using Adobe Photoshop (Adobe Inc., San Jose, CA, USA). Karyotype analysis and identification of individual chromosomes with fluorescent signals were performed according to bulb onion nomenclature [18] and previously published karyotypes of closely related *Allium* species [19]. The relative position of the recombination site on the relevant chromosome arm was a ratio between its arm length and distance from centromere to the recombination point.

### 2.7. Cytoplasmic Evaluations

Genomic DNA was isolated from the pooled leaf tissue of all accessions and cytoplasms were classified using high-resolution melting (HRM) of an indel in the chloroplast accD gene [20]. Controls included previously isolated genomic DNA from *A. cepa* (N-cytoplasmic inbred B1750B and S-cytoplasmic B1750A), *A. fistulosum*, and *A. galanthum*.

### 2.8. Stemphyllium Screening

The AVON accessions listed in Table 1 were developed in fields under natural heavy disease pressure at the World Vegetable Center in Shanhua, Taiwan. Seven of the accessions (AVON 1275, 1278, 1282, 1283, 1284, 1290, and 1291), the cultivar ‘Green Banner’ (*A. fistulosum*), and four cultivars of bulb onion were evaluated for SLB in greenhouses/mist chambers at Seminis Vegetable Seeds in Deforest, WI. Plants were grown in Fafard 4P^®^ mix (Sun Gro Horticulture, Agawam, MA, USA) in trays with 5-cm cells in a greenhouse at 24 to 27 °C with 12 h lights. Four replications of 18 plants of each accession were sprayed three times at 8, 9, and 10 weeks after sowing with 1 × 10^5^ conidia of *S. vesicarium* per mL in 0.01% tween-20. Plants were then placed into a mist chamber to maintain leaf wetness for 48 h, and then returned to the greenhouse. Eight days after the third inoculation, disease severity in each replication was visually rated by four individuals using a scale of 1 to 5, where 1 = no symptoms to 5 = severe leaf blight or dead plants. Disease severity ratings were averaged over replications. RStudio was used for statistical analyses [21]. One-way ANOVA was calculated based on four replications of each accession, and the least significant differences were calculated using RStudio.

## 3. Results

### 3.1. Stemphylium Leaf Blight Evaluations

During development, the interspecific hybrid, derived accessions (Table 1), and cultivars of *A. fistulosum* and *A. cepa* were planted in the field at the World Vegetable Center (Shanhua, Taiwan) and subjected to natural disease pressure by *S. vesicarium*. In these fields, AVON 1275 and derived accessions (Table 1), as well as the original *A. fistulosum* parent (AF468), appeared resistant to SLB, while all cultivars of *A. cepa* appeared susceptible. In greenhouse evaluations, there were significantly different (*P* < 0.001) disease severity reactions (DSRs), which ranged from 1.3 to 2.3 for *A. fistulosum* and derived accessions, whereas cultivars of *A. cepa* had significantly higher DSRs ranging from 2.8 to 4.6 (Table 2). Because *A. fistulosum*, AVON 1275, and many derived lines had significantly higher resistance to SLB than *A. cepa*, resistance from *A. fistulosum* appears to show some level of dominance over susceptibility of the bulb onion (Table 2).

### 3.2. GISH Analysis

#### 3.2.1. Chromosome Complement of Amphidiploid S_5_ Generation

The F_1_ interspecific hybrid was self-pollinated for five generations (S_5_) and GISH evaluation of all progenies (AVON 1275) showed that the all hybrids were allotetraploids (2*n* = 4*x* = 32) possessing 16 chromosomes from *A. cepa* (16C) and 16 chromosomes from *A. fistulosum* (16F) (Figure 1a). No recombination was detected between the chromosomes of *A. cepa* and *A. fistulosum*. Karyotype analysis (Figure 1a’) confirmed the presence of complete diploid sets of both parental species: 2*n* = 4*x* = 32 (16C + 16F).

#### 3.2.2. Chromosome Complement of Amphidiploid Backcross Progenies

Amphidiploid AVON 1275 was backcrossed as the female parent with *A. cepa*. GISH analysis of AVON 1278 (S_5_BC_1_) showed that the hybrids were allotetraploid with 16 chromosomes of *A. cepa* and 16 chromosomes of *A. fistulosum* (Figure 1b) and no recombinant chromosomes. Karyotype analysis (Figure 1b’) showed the presence of complete diploid sets of both parental species: 2*n* = 4*x* = 32 (16C + 16F).

GISH analysis of AVON 1282, 1290, 1501, and 1502 (all S_4_BC_1_ families) showed that the hybrids possessed 16 chromosomes of *A. cepa* and 16 chromosomes of *A. fistulosum* (Figure 1c–f) with no recombinant chromosomes. Karyotype analysis (Figure 1c’–f’) showed the presence of complete diploid sets of both parental species: 2*n* = 4*x* = 32 (16C + 16F). These results indicate that progenies were not derived from backcrossing to bulb onion, but from self-pollination of the amphidiploid parent.

#### 3.2.3. Backcross Progenies with Missing Chromosomes

GISH analysis of AVON 1283 (S_4_BC_1_) showed that plants possessed 30 chromosomes, 16 chromosomes from *A. cepa* and 14 from *A. fistulosum* (Figure 2a). Karyotype analysis (Figure 2a’) revealed the presence of a complete diploid set of chromosomes from *A. cepa*, and seven pairs of chromosomes from *A. fistulosum*, with one pair of chromosome 5 missing (2*n* = 4*x* − 2 = 30) (16C + 14F). GISH analysis of AVON 1284 (S_3_BC_1_) revealed that the hybrid also contained 30 chromosomes (2*n* = 4*x* − 2 = 30) and karyotype analysis (Figure 2b,b’) showed the presence of a complete diploid set of chromosomes of *A. cepa* and seven pairs from *A. fistulosum*, also missing the pair of homologous chromosomes 5 (16C + 14F).

#### 3.2.4. Backcross Progenies with Recombinant Chromosomes

GISH analysis of AVON 1291 (S_4_BC_1_) showed that the hybrids were allotetraploid, possessing 16 chromosomes from *A. cepa*, 15 chromosomes from *A. fistulosum*, and one recombinant chromosome between *A. cepa* and *A. fistulosum* (2*n* = 4*x* = 32) (Figure 3a). Karyotype analysis (Figure 3a’) showed the presence of complete diploid set of *A. cepa* (8 pairs of homologous chromosomes), 7 pairs of *A. fistulosum* homologous chromosomes, and one pair of *A. fistulosum* chromosome 2, which contained one recombinant chromosome with a centromeric region belonging to *A. fistulosum* and one non-recombinant chromosome (16C + 15F + 1F/C). The recombination site was located on the long arm of chromosome 2. The relative position of recombination site from the centromere was 83.6% ± 0.6 (SD, *n* = 8). GISH analysis of AVON 1504 (S_4_BC_1_) showed that the hybrids were allotetraploids possessing 16 chromosomes from *A. fistulosum*, 14 chromosomes from *A. cepa*, and 2 recombinant chromosomes between *A. cepa* and *A. fistulosum* (2*n* = 4*x* = 32) (Figure 3b). Karyotype analysis (Figure 3b’) showed the presence of complete diploid sets of *A. fistulosum*, 7 pairs of the *A. cepa* homologous chromosomes and one pair of recombinant chromosome 7 with the centromeric region from *A. cepa* (16F + 14C + 2C/F). The recombination site is located in the short arm of chromosome 7, and the relative positions of recombination sites on both chromosomes were the same at 49.7% ± 0.3 (SD, *n* = 5). These results clearly demonstrate recombination between chromosomes from *A. cepa* and *A. fistulosum*, supporting larger backcrossing efforts towards introgression of beneficial traits into the bulb onion.

### 3.3. Pollen Fertility

Pollen fertility of the advanced generations from interspecific hybrids between *A. cepa* and *A. fistulosum* was evaluated using acetocarmine stained pollen. The plants were male−fertile, with fertility ranging from 60 to 95% (Table 3). Plants from AVON 1291 which possessed one recombinant chromosome 2 showed 93% stainable pollen, revealing that the recombinant chromosome did not significantly reduce male fertility.

### 3.4. Cytoplasms of Plants Derived from Interspecific Crosses between A. cepa and A. fistulosum

Cytoplasms of plants from advanced generations from interspecific hybrids were determined using high-resolution melting (HRM) of an indel in the chloroplast accD gene [20]. This marker distinguishes normal (N) male−fertile cytoplasm from male−sterile (S) cytoplasm of onion, as well as the cytoplasms of *A. fistulosum* and *A. galanthum* (Figure 4). None of the accessions originating from the original interspecific cross possessed the cytoplasm of *A. fistulosum*. Most interspecific hybrids between *A. cepa* and *A. fistulosum* have been produced using *A. fistulosum* as the female parent, which limits introgression due to deleterious nucleo-cytoplasmic interactions [22]. In theory, chromosome regions carrying disease or pest resistance(s) in these accessions should avoid this deleterious interaction because they possess N cytoplasm of *A. cepa.*

## 4. Discussion

GISH clearly distinguished chromosomes from *A. cepa* and *A. fistulosum* despite the presence of similar DNA sequences in their genomes, such as a subtelomeric tandem repeat showing 80% identity [23,24]. Additionally, retrotransposons are abundant in the *Allium* genomes [25] and are dispersed along entire chromosomes, which may contribute to cross-hybridization. GISH analyses of interspecific hybrids between *A. cepa* and *A. fistulosum* revealed that spontaneous chromosome duplication produced allotetraploids. These results agree with Levan [26], who reported polyploid F_2_ plants from crosses between *A. cepa* and *A. fistulosum* without artificial chromosome duplication. S_5_ plants of AVON1275 were allotetraploids with complete diploid chromosome sets from both parents and were used as the female parent for backcrossing with *A. cepa*. Most of the progenies from the backcross generations were allotetraploids (2*n* = 4*x* = 32) with complete diploid sets from both parental species (16C + 16F). Equal chromosome numbers from both parents indicates that progenies were from self-pollination and not from backcrosses to *A. cepa*. The same observation was reported by Budylin et al. [27] who completed GISH analyses of advanced generations of hybrids between *A. cepa* and *A. fistulosum* showing resistance to downy mildew. These authors revealed spontaneous polyploidization and absence of predominantly *A. cepa* chromosomes in backcross generations. The absence of backcrossing in diploid F_1_BC_3_ hybrids between *A. cepa* and *A. fistulosum* was also reported by Hou and Peffley [10]. Our observations indicate that the amphidiploid plants of S_4_BC_1_ lines AVON 1282, 1290, 1501, and 1502 were derived from the selfing of the amphidiploid female parent as opposed to backcrossing.

Plants from AVON 1283 (S_4_BC_1_) and 1284 (S_3_BC_1_) possessed a complete diploid set of chromosomes from *A. cepa* and seven pairs of *A. fistulosum* chromosomes, with one pair of chromosome 5 missing (16C + 14F − 2Ch5). Because both AVON 1283 and 1284 are resistant to SLB, this indicates that SLB resistance from *A. fistulosum* is not carried on chromosome 5.

Plants from AVON 1291 (S_4_BC_1_) and AVON 1504 (S_4_BC_1_) possessed recombinant chromosomes. In AVON1291, one recombinant chromosome 2 was observed and its homolog was non-recombinant chromosome 2 of *A. cepa.* It is likely that a crossing over event occurred between *A. cepa* and *A. fistulosum* chromosomes at the allotetraploid stage. Homeologous recombination has been detected in both recently formed natural polyploids and synthetic interspecific polyploids [28]. Recombination between chromosomes of *A. cepa* and *A. fistulosum* has been documented [10,13,29], and intergenomic pairing and recombination have been documented for allotetraploid hybrids of sour cherry (*Prunus cerasus*), *Solanum lycopersicum* × *Solanum lycopersicoides*, and for allopolyploids in asterids [30]. Two recombinant chromosome 7 were revealed in AVON 1504 with the centromeric region belonging to *A. cepa.* The size of the *A. cepa* region within both recombinant chromosomes 7 was the same, at 49.7% ± 0.3 (SD, *n* = 5), which indicates the presence of two sister chromatids that did not segregate during the second meiotic division and remained in the same gamete. This suggests that spontaneous polyploidization in the interspecific hybrids between *A. cepa* and *A. fistulosum* was caused by second division restitution, resulting in 2*n*-gamete formation. Alternatively, the presence of two identical recombinant chromosomes 7 could be the result of the somatic doubling of the complement in the zygote. The recombinant chromosome could have been formed during meiosis in the interspecific hybrid, either in a pollen mother cell or in a macrospore mother cell and underwent somatic doubling. Another interesting observation from the GISH analysis of AVON 1504 was the lack of genetic material from *A. cepa* in the distal region on the short arm of the recombinant chromosomes 7, which has been replaced with genetic material of *A. fistulosum* (Figure 3b,b’,b”). It is known that 5S rDNA is located on the short arm of chromosome 7 in both species [31,32] and the lengths and sequences of the coding 5S rRNA gene segments are conserved among *Allium* species [33]. Based on this observation, we conclude that this part of the *A. cepa* genome can be replaced by *A. fistulosum* and function in the cytoplasm of *A. cepa*.

Spontaneous chromosome doubling in interspecific hybrids between *A. cepa* and *A. fistulosum* was observed in our materials and has been previously reported [26]. This could be caused by fusion of 2n gametes, what led to the formation of allotetraploid progeny possessing complete diploid sets from both parental species. However, the formation of restitution gametes is rare in *A. cepa* [34]. Even if we assume the formation of restitution gametes, the probability of fusion of 2*n*-egg with 2*n*-sperm is unlikely. An alternative hypothesis may be that facultative apomixes occurred, which has been reported for several species of genus *Allium* [35,36]. Spontaneous polyploidization could be also caused by cytomixis, which involves the transfer of nuclear material from one cell to another via cytomictic channels [37]. Cytomixis has been observed in the pollen mother cells of *A. cepa* and *A. fistulosum* [38,39].

## 5. Conclusions

Spontaneous polyploidization and no evidence of backcrossing were revealed in backcross progenies from interspecific hybrids between *A. cepa* and *A. fistulosum*. The mechanism of spontaneous polyploidization is not clear, and it will be of great interest to determine if it results from somatic or meiotic chromosome doubling. Clear evidence for recombination between the chromosomes of *A. cepa* and *A. fistulosum* supports continued research to introgress beneficial traits, such as SLB resistance, from *A. fistulosum* into the bulb onion.

## Figures and Tables

**Figure 1 genes-10-00195-f001:**
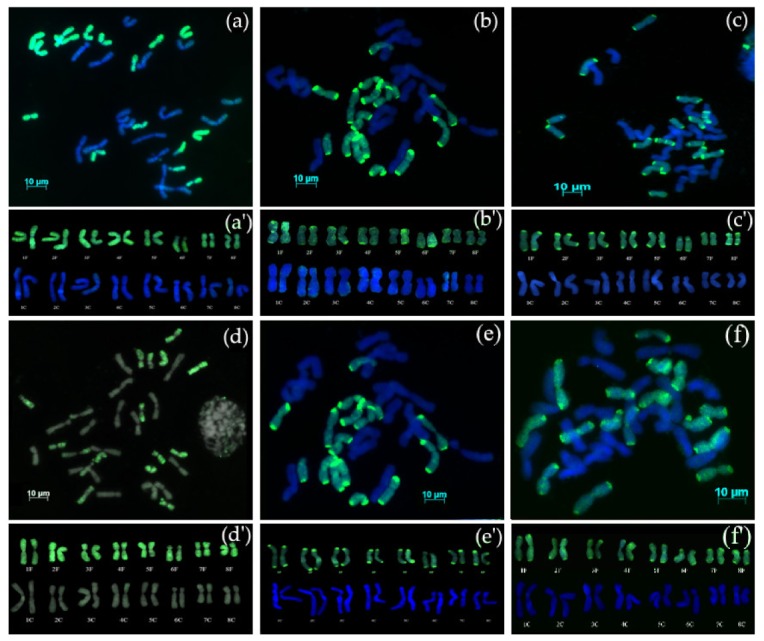
Genomic in situ hybridization (GISH) on mitotic metaphase chromosomes of advanced generations from interspecific hybrids between *A. cepa* and *A. fistulosum*. (**a**) AVON 1275 (S_5_), 2*n* = 4*x* = 32, with 16 chromosomes of *A. cepa* and 16 chromosomes of *A. fistulosum*; (**b**) AVON 1278 (BC_1_), 2*n* = 4*x* = 32, with 16 chromosomes of *A. cepa* and 16 chromosomes of *A. fistulosum*; (**c**) AVON 1282 (S_4_BC_1_) 2*n* = 4*x* = 32, with 16 chromosomes of *A. cepa* and 16 chromosomes of *A. fistulosum*; (**d**) AVON 1290 (S_4_BC_1_), 2*n* = 4*x* = 32, with 16 chromosomes of *A. cepa* and 16 chromosomes of *A. fistulosum*; (**e**) AVON 1501 (S_4_BC_1_), 2*n* = 4*x* = 32, with 16 chromosomes of *A. cepa* and 16 chromosomes of *A. fistulosum*; (**f**) AVON 1502 (S_4_BC_1_), 2*n* = 4*x* = 32, with 16 chromosomes of *A. cepa* and 16 chromosomes of *A. fistulosum*; (**a’**, **b’**, **c’**, **d’**, **e’**, **f’**) karyotypes of the AVON1275, AVON121278, AVON 1282, AVON 1290, AVON 1501, and AVON1502 hybrids, respectively. Digoxigenin labelled probe DNA of *A. fistulosum* and detected with FITC (green fluorescence).

**Figure 2 genes-10-00195-f002:**
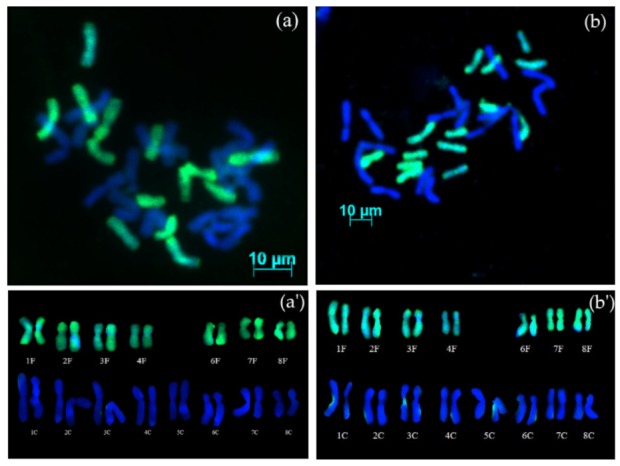
GISH on mitotic metaphase chromosomes of advanced generations from interspecific hybrids between *A. cepa* and *A. fistulosum*: (**a**) AVON 1283 (S_4_BC_1_) contains 16 chromosomes of *A. cepa* and 14 chromosomes of *A. fistulosum*; (**b**) AVON 1284 (S_3_BC_1_) contains 16 chromosomes of *A. cepa* and 14 chromosomes of *A. fistulosum*; (**a’,b’**) karyotypes of the AVON 1283 and AVON 1284 hybrids, respectively, with missing one pair of homologous chromosomes 5 of *A. fistulosum*.

**Figure 3 genes-10-00195-f003:**
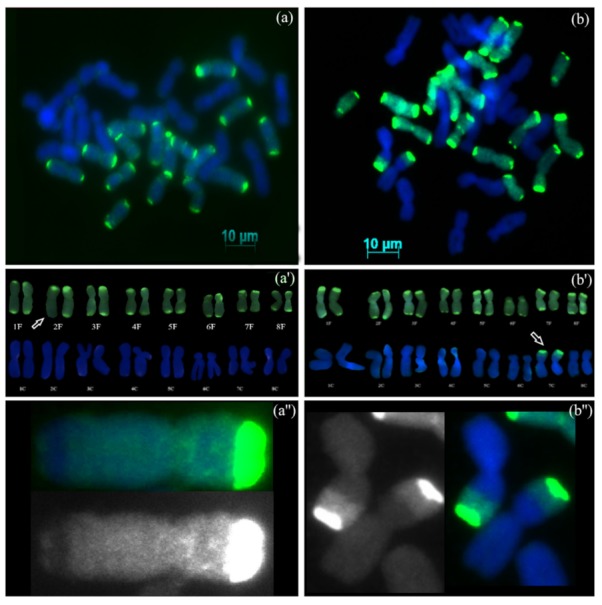
GISH on mitotic metaphase chromosomes of advanced generation progenies from interspecific hybrids between *A. cepa* and *A. fistulosum*. (**a**) AVON 1291, (S_4_BC_1_) with complete diploid set of *A. cepa*, and 7 pairs of *A. fistulosum* homologous chromosomes and one homologous chromosome 2 of *A. fistulosum*, and another one was recombinant; (**b**) AVON 1504 (S_4_BC_1_) with complete diploid sets of *A. fistulosum*, 7 pairs of *A. cepa* homologous chromosomes and two recombinant chromosomes 7; Digoxigenin labelled probe DNA of *A. fistulosum* and detected with FITC (green fluorescence); (**a’**), karyotype of the AVON1291 hybrid with one recombinant chromosome 2; (**b’**) karyotype of the AVON1504 hybrid with two recombinant chromosomes 7; (**a”**) recombinant chromosome 2: top — the merged DAPI and FITC filter images, below — the FITC filter image; (**b”**) two recombinant chromosomes 7: left — the FITC filter image, right — the merged DAPI and FITC filter image; arrows indicate recombinant chromosomes.

**Figure 4 genes-10-00195-f004:**
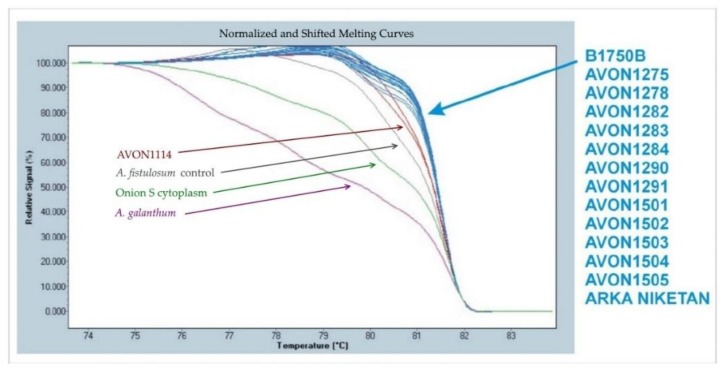
High resolution melting (HRM) analysis of the accD indel distinguishing normal (N) male−fertile cytoplasm of onion (blue) from the cytoplasms of male−sterile (S) onion (green), *Allium fistulosum* (red) and *Allium galanthum* (violet). Lines from more than one DNA may plot together.

**Table 1 genes-10-00195-t001:** Parental *Allium* species and advanced generations derived from interspecific hybrids used for cytogenetic studies.

Accession	Pedigree	Generation
AVON 1275	TA207 × AF468	S_5_
AVON 1278	AVON 1275 × AC464	S_5_BC_1_
AVON 1284	AVON 1275 × AC464	S_3_BC_1_
AVON 1282	AVON 1275 × AC464	S_4_BC_2_
AVON 1283	AVON 1275 × AC464	S_4_BC_2_
AVON 1290	AVON 1275 × AC464	S_4_BC_2_
AVON 1291	AVON 1275 × AC464	S_4_BC_2_
AVON 1501	AVON 1275 × AC464	S_4_BC_2_
AVON 1502	AVON 1275 × AC464	S_4_BC_2_
AVON 1503	AVON 1275 × AC464	S_4_BC_2_
AVON 1504	AVON 1275 × AC464	S_4_BC_2_
AF468	*Allium fistulosum* ‘Tuwel’	
AC464	*Allium cepa* ‘Arka Niketan’	
TA207	*Allium cepa* ‘Rouge de Tana’	

**Table 2 genes-10-00195-t002:** Mean disease severity ratings (DSRs) ± standard deviations (sd) for reactions to *Stemphyllium vesicarium* by *Allium fistulosum*, *A. cepa*, and accessions derived from AVON 1275.

Accession ^1^	Cultivar	Mean DSR ± SD ^2^
*A. fistulosum*	Green Banner	1.3 ± 0.5 a
AVON 1282		1.3 ± 0.3 a
AVON 1278		1.6 ± 0.5 ab
AVON 1283		1.6 ± 0.6 ab
AVON 1291		1.8 ± 0.5 ab
AVON 1284		2.0 ± 0.4 ab
AVON 1290		2.0 ± 0.7 ab
AVON 1275		2.3 ± 0.9 abc
*A. cepa*	1620	2.8 ± 0.6 abc
*A. cepa*	1606	3.1 ± 0.6 bcd
*A. cepa*	Granex 33	3.8 ± 0.9 cd
*A. cepa*	1707	4.6 ± 0.5 d

^1^ Origins of AVON accession are listed in Table 1. Accessions of *A. cepa* are proprietary lines of Seminis Seed Company (Woodland, CA, USA); ^2^ DSR of 1 = no symptoms and 5 = very severe blight or dead plants. Means followed by the same letter were not significantly different based on least significant difference at P = 0.5.

**Table 3 genes-10-00195-t003:** Pollen fertility (percentage of dark red stained pollen grains) of advanced generations of interspecific hybrids between *A. cepa* and *A. fistulosum.*

Accession	Pollen Fertility, %
AVON 1275 (S_5_)	94.1
AVON 1290 (S_4_BC_2_)	85.3
AVON 1291 (S_4_BC_2_)	92.9
AVON 1501 (S_4_BC_2_)	63.1
AVON 1502 (S_4_BC_2_)	94.7

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
