# Peer review of "Cytological Evaluations of Advanced Generations of Interspecific Hybrids between Allium cepa and Allium fistulosum Showing Resistance to Stemphylium vesicarium"

_genes, 2019, doi:10.3390/genes10030195_

Round 1

Reviewer 1 Report

I had read this manuscript with a great interest. This manuscript deals with a visualization of chromosomal constitutions of advanced generations of Allium interspecific hybrids by means of GISH technique. The contents of this study could not be worthwhile for publishing in the present condition of the manuscript. However, a better way would be to reconsider the following major and minor points in the manuscript before re-evaluation.

(Major points)

1. Recommendation of an important reference

SABRAO Journal of Breeding and Genetics, 36(2) 107-112, 2004

CHARACTERIZATION OF AN INTERSPECIFIC CROSS BETWEEN JAPANESE BUNCHING ONION (Allium fistulosum) AND ONION (A. cepa)

This paper explained the original parents and F1 hybrid of the advanced generation in this manuscript. To ensure the pedigree of the generation, the authors should read and understand more detailed contents of this previous study.

2. The necessity of more explanation on disease resistant test

The description of inoculation test for Stemphylium leaf blight is not find in all the parts of this manuscript including M&M, Results and Discussion. Especially, the result of disease severity test should be included with the numerical data in each hybrid lines as well as parental lines. This is a fatal flaw for this manuscript.

3. The credibility of backcross progeny

In all of the backcross progeny listed in Table 1, the chromosome numbers were 2n=30 or 32. If the backcrossing was successful, the number should be 2n=24. Actually, this is another fatal flaw of this study. Perhaps, an explanation for the failure of backcrossing is not reasonable way to interpret this case. Anyway, the reviewer feels less credibility of these breeding lines.

(Minor points)

P1L38: Allium -> should be Italic

P2L57: establish -> confirm

P2L71: self-crossing -> selfing or self-pollination

P3L98: Allium -> should be Italic

P3L98-100: The sentence should be modified like this;

The relative position of recombination site in the relevant chromosome arm was a ratio between its arm length and a distance from centromere to crossing point.

->-> How many cells did you observe to ensure the results?

P3L102: The sentence should be modified like this;

DNA was isolated from the pooled leaf tissue of all accessions, and ~~~

P4L112: Allium -> should be Italic

P4L119: (16A) -> (16C), (16F; Figure 1a) -> (16F) (Figure 1a)

P4L121: 4n=2C=32 -> 2n=4x=32 *n and x should be Italic.

P4L126: 4n=2C=32 (16A+16F) -> 2n=4x=32 (16C+16F) *n and x should be Italic.

P4L127: The sentence should be modified like this;

~~~ lines, AVON 1282, 1290, 1501 and 1502 showed that ~~~~

P4L131: 4n=2C=32 (16A+16F) -> 2n=4x=32 (16C+16F) *n and x should be Italic.

P4L136-137: 4n-2=2C=30 (16A+14F) -> 2n=4x-2=30 (16C+14F) *n and x should be Italic.

P4L141-142: 4n-2=2C=30 (16A+14F) -> 2n=4x-2=30 (16C+14F) *n and x should be Italic.

P5L145-150: 4n=2C=32 -> 2n=4x=32 *n and x should be Italic.

P6L164-165: 4n=2C=32 (16A+15F+1F/A) -> 2n=4x=32 (16C+15F+1F/C) *n and x should be Italic.

P6L166: 83.6% ± 0.04 -> 83.6 ± 0.04% (mean ± SD or SE?, n=?)

P6L171-172: 4n=2C=32 (16F+14A+2F/A) -> 2n=4x=32 (16F+14C+2C/F) *n and x should be Italic.

P6L173: 49.7% ± 0.05 -> 49.7 ± 0.05% (mean ± SD or SE?, n=?)

P7183-186: Please show readers individual data on the pollen fertilities of all plant materials including parental lines.

P9L208: 4n=2C=32 (16A+16F) -> 2n=4x=32 (16C+16F) *n and x should be Italic.

P9L214: Allium -> should be Italic

P9L223: in backcrossing generations -> in backcross progeny

P9L237: Allium -> should be Italic

P9L247: Allium -> should be Italic

Tables 1 and 2:

For AVON1503, the result of GISH could not be found.

For AVON1505, an information on plant species is lacking.

No. lot, CepaxFistulosum no., and pedigree seem to be internal information in WVC. No one understand these meaning. Please indicate the meaning of these code.

Fig. 3: Please explain the roles of arrows which indicate the chromosomes.

Author Response

Reviewer 1

1. Recommendation of an important reference

SABRAO Journal of Breeding and Genetics, 36(2) 107-112, 2004

CHARACTERIZATION OF AN INTERSPECIFIC CROSS BETWEEN JAPANESE BUNCHING ONION (Allium fistulosum) AND ONION (A. cepa)

This paper explained the original parents and F1 hybrid of the advanced generation in this manuscript. To ensure the pedigree of the generation, the authors should read and understand more detailed contents of this previous study.

Thank you for your recommendation and providing the reference to Singh et al. 2004.  We carefully read this paper.  We carried out the GISH analysis of hybrids obtained by crossing A. cepa ‘Arka Niketan’ (TA 420) and A. cepa TA 207 as a female parent and A. fistulosum as a male parent. In the above mentioned paper, the authors used for crossing Japanese bunching onion (VRJBO-77) as the female parent and A. cepa cv. Agri Dark Red as a male parent. These interspecific hybrids will possess the cytoplasm of A. fistulosum.  In our study, the interspecific hybrid possess normal male-fertile cytoplasm of onion, and were backcrossed with A. cepa.  Therefore we feel that this paper is not salient to our research.

2. The necessity of more explanation on disease resistant test

The description of inoculation test for Stemphylium leaf blight is not find in all the parts of this manuscript including M&M, Results and Discussion. Especially, the result of disease severity test should be included with the numerical data in each hybrid lines as well as parental lines. This is a fatal flaw for this manuscript.

Description of the disease evaluation and results are presented in our revision. 

3. The credibility of backcross progeny

In all of the backcross progeny listed in Table 1, the chromosome numbers were 2n=30 or 32. If the backcrossing was successful, the number should be 2n=24. Actually, this is another fatal flaw of this study. Perhaps, an explanation for the failure of backcrossing is not reasonable way to interpret this case. Anyway, the reviewer feels less credibility of these breeding lines.

This is one of our main points in that breeders must careful and not carry forward self pollinations of the amphidiploid or diploid backcross parent.  Nevertheless we document that crossing over had occurred and it should be possible to transfer chromosome regions from A. fistulosum to bulb onion, and that breeders should use cytogenetic evaluations in the beginning of the interspecific breeding and during backcrossing.

(Minor points)

All corrections were made in the revision:

P1L38: Allium -> should be Italic

P2L57: establish -> confirm

P2L71: self-crossing -> selfing or self-pollination

P3L98: Allium -> should be Italic

P3L98-100: The sentence should be modified like this;

The relative position of recombination site in the relevant chromosome arm was a ratio between its arm length and a distance from centromere to crossing point.

->-> How many cells did you observe to ensure the results?

P3L102: The sentence should be modified like this;

DNA was isolated from the pooled leaf tissue of all accessions, and ~~~

P4L112: Allium -> should be Italic

P4L119: (16A) -> (16C), (16F; Figure 1a) -> (16F) (Figure 1a)

P4L121: 4n=2C=32 -> 2n=4x=32 *n and x should be Italic.

P4L126: 4n=2C=32 (16A+16F) -> 2n=4x=32 (16C+16F) *n and x should be Italic.

P4L127: The sentence should be modified like this;

~~~ lines, AVON 1282, 1290, 1501 and 1502 showed that ~~~~

P4L131: 4n=2C=32 (16A+16F) -> 2n=4x=32 (16C+16F) *n and x should be Italic.

P4L136-137: 4n-2=2C=30 (16A+14F) -> 2n=4x-2=30 (16C+14F) *n and x should be Italic.

P4L141-142: 4n-2=2C=30 (16A+14F) -> 2n=4x-2=30 (16C+14F) *n and x should be Italic.

P5L145-150: 4n=2C=32 -> 2n=4x=32 *n and x should be Italic.

P6L164-165: 4n=2C=32 (16A+15F+1F/A) -> 2n=4x=32 (16C+15F+1F/C) *n and x should be Italic.

P6L166: 83.6% ± 0.04 -> 83.6 ± 0.04% (mean ± SD or SE?, n=?)

P6L171-172: 4n=2C=32 (16F+14A+2F/A) -> 2n=4x=32 (16F+14C+2C/F) *n and x should be Italic.

P6L173: 49.7% ± 0.05 -> 49.7 ± 0.05% (mean ± SD or SE?, n=?)

P7183-186: Please show readers individual data on the pollen fertilities of all plant materials including parental lines.

P9L208: 4n=2C=32 (16A+16F) -> 2n=4x=32 (16C+16F) *n and x should be Italic.

P9L214: Allium -> should be Italic

P9L223: in backcrossing generations -> in backcross progeny

P9L237: Allium -> should be Italic

P9L247: Allium -> should be Italic

Tables 1 and 2:

For AVON1503, the result of GISH could not be found.

Seeds of AVON1503 did not germinate. We removed this accession from the table.

For AVON1505, an information on plant species is lacking.

We added the information to the table.

No. lot, CepaxFistulosum no., and pedigree seem to be internal information in WVC. No one understand these meaning. Please indicate the meaning of these code.

We adapted the table.

Fig. 3: Please explain the roles of arrows which indicate the chromosomes.

We added to Figure legend the explanation for arrows.

Reviewer 2 Report

Interspecific crossing is a promising approach for introgression of valuable traits to  develop cultivars with improved characteristics. Allium fistulosum L. possesses numerous pest  resistances that are lacking in Allium cepa L., including resistance to Stemphylium leaf blight (SLB).  Genomic in situ hybridization (GISH) was used to study the chromosomal composition of advanced  generations of interspecific hybrids between A. cepa and A. fistulosum which showed resistance to  SLB.

There are the following questions:

1 How to explain the differences among the results of GISH test on 4 individual of BC2F4 strains of backcross generation? AVON1282 and 1501 presented signals only on the ends of chromosomes, while AVON1290 and 1502 presented signals on almost the whole chromosome.

2 How to determine the deletion of a pair of NO.5chromosome in AVON1283 and 1284? Have they been detected with molecular markers specific to NO.5 chromosome?

Author Response

Reviewer 2

Interspecific crossing is a promising approach for introgression of valuable traits to  develop cultivars with improved characteristics. Allium fistulosum L. possesses numerous pest  resistances that are lacking in Allium cepa L., including resistance to Stemphylium leaf blight (SLB).  Genomic in situ hybridization (GISH) was used to study the chromosomal composition of advanced  generations of interspecific hybrids between A. cepa and A. fistulosum which showed resistance to  SLB.

There are the following questions:

1 How to explain the differences among the results of GISH test on 4 individual of BC2F4 strains of backcross generation? AVON1282 and 1501 presented signals only on the ends of chromosomes, while AVON1290 and 1502 presented signals on almost the whole chromosome.

We replaced 1501, 1283 images for images with more clear hybridization signals along entire chromosome. Concerning pronounced signal at the distal end of the A. fistulosum chromosomes: It is known that most members of the Allium genus have a 375 bp common subtelomeric repeat (Pich et al. 1996). However, based on the GISH result we assume that A. fistulosum has species specific repeat in addition to common subtelomeric repeat, its own specific subtelomeric repeat, which gives such a strong signal.

2 How to determine the deletion of a pair of NO.5chromosome in AVON1283 and 1284? Have they been detected with molecular markers specific to NO.5 chromosome?

Chromosome 5 of A. fistulosum in hybrids can be identified by GISH and chromosome morphology – metacentric chromosome with relative length -11.5 ±1.5 and centromere index 47.9±1.9.  We are presently working to develop molecular markers to distinguish A. cepa chromosomes from A. fistulosum, which will aid in the future to identify chromosomes. 

Reviewer 3 Report

The paper describes the staining of chromosomes in progeny plants of onion and fistulosum using GISH, which is a delicate method to study chromosomal behaviour in plants resulting from crosses between plants belonging to different species. The paper is describing an interesting phenomenon and can definitely become a very nice paper, but it now needs further improvement before it can be published. Recommendations are presented below.

Introduction

Ø  In the past, the last author of this paper was involved in GISH studies analysing crosses between A. cepa and the F1(A. roylei x A. fistulosum). These crosses were made to circumvent problems with sterility in progeny plants obtained from crosses between cepa and fistulosum. It was shown in that paper that by following this approach introgression of traits from fistulosum into cepa could take place. It is important to clarify in the introduction, why the authors decided to study direct crosses between A. cepa and A. fistulosum, knowing the problems with sterility that may occur in F1 progeny plants (and also of chromosome doubling by papers from other authors) and knowing of an alternative solution.

Ø  Line 54: only across for chromosomal regions. Please leave out the word ‘for’

Material and Methods

Ø  Line 68: An overview of the parental species, their hybrid (F5) that was used for backcrossing, and advanced generations of interspecific hybrids are given in Table 1.

Remark: This is confusing as according to the Table also F3 and F4 hybrid plants were used for backcrossing? Please adapt.

Table 1. suggests the presence of plants in BC1F5 generation, BC2F5 generation and also BC2F4 and BC1F3 generations. Based on the GISH results it is clear that such plants did clearly not exist, as the chromosome number should be lower (eg 24) in case F5 plants were back crossed with onion plants. Since this is not the case in all but two genotypes, it is more likely that these plants occurred after apomixis or were selfings from AVON1275.

Ø  Line 78: Others hybrid lines did not flowering under our greenhouse conditions.

Change into ‘flower’

Ø  Line 98: as those previously described

Leave out ‘those’

Ø  Line 108   We were able with GISH clearly distinguished the parental genomes

Change into We were able with GISH to clearly distinguish

Ø  Linea 109-112: ...despite the presence of highly identical DNA sequences in their genomes. It is known that A. cepa and A. fistulosum possessed subtelomeric tandem repeats with 80% of identity [20,21]. Also, retrotransposons are abundant in the Allium genomes [22], which are dispersed along entire chromosomes and may cause a probe in situ cross-hybridization.

Please transfer these lines 109-112 to the discussion. This is not part of the results.

Ø  Line 113-115: In the hybridization mixture, we used a ratio of 1: 30 of probe and block DNA, and a 78% stringency washing was applied. GISH analyses were performed on 3-5 plants from each line of advanced interspecific hybrids.

This information belongs to the M&M

Ø  Line 117: Leave out this part ‘which were used in backcrossing as a female parent,’ as it is not informative at that place

Ø  It was not revealed recombinant chromosomes between A. cepa and A. fistulosum.

Please adapt this sentence

Ø  Line 247: This study demonstrated that GISH is a powerful tool for identification of parental genomes at the chromosomal level in Allium interspecific hybrids.

This has been shown before by Khrustaleva & Kik, not new.

Ø  Line 248: The spontaneous diploidization and absence of introgression into A. cepa during back-crossing were revealed in BC1F3, BC1F5, BC2F4 accessions.

There is no absence of introgression, but results show that there is no evidence of back crossing! It is highly likely that in most plants apomixis has occurred (or selfing and then indeed without recombination).

Ø  Line 249: We suggested that diploidization caused by 2n-gamete formation due to second division restitution in meiosis.

Explain: How would this be a matter of 2n-gamete formation? Both in the mother and in the father, meaning that both the egg cells and the pollen cells have a double number of chromosomes? Could this polyploidization not have happened after formation of the F1-zygote?

Ø  Line 250: Cytoplasmic determination showed the presence of bulb onion N cytoplasm in all analyzed hybrids. Our evaluation of both the nuclear and cytoplasm inherited genetic material makes very promising the use of the analyzed accessions in onion breeding.

Explain: Why is this of importance? Why is TA207 not included in the HRM analysis? The first cross was between TA207 and AF468, which finally resulted in AVON1275. So AVON1275 is expected to have the same cytoplasm as TA207. So, that all ‘back cross’ plants have the same cytoplasm as  progeny using AVON1275 is fully as expected. How does this result contribute to the paper or should it be discarded?

Add also information on how this result can be of interest to onion breeding and what needs to be done in the first place (haploidization).

Figures are not always clear explained: why are some genotypes stained with Cy3 and others with FITC? This should be described in the M&M. It is also not clear why in some genotypes especially the telomeres are more stained than the rest of the chromosomes. In some genotypes it almost looks as if only the telomers are stained green and not the rest of the chromosomes. More explanation is desired.   

In the Discussion, attention should also be given to the fact that in AVON1291 the recombinant chromosome is in a homozyogus state (when did this occur?). 

Author Response

Reviewer 3

The paper describes the staining of chromosomes in progeny plants of onion and fistulosum using GISH, which is a delicate method to study chromosomal behaviour in plants resulting from crosses between plants belonging to different species. The paper is describing an interesting phenomenon and can definitely become a very nice paper, but it now needs further improvement before it can be published. Recommendations are presented below.

Introduction

In the past, the last author of this paper was involved in GISH studies analysing crosses between A. cepa and the F1(A. roylei x A. fistulosum). These crosses were made to circumvent problems with sterility in progeny plants obtained from crosses between cepa and fistulosum. It was shown in that paper that by following this approach introgression of traits from fistulosum into cepa could take place. It is important to clarify in the introduction, why the authors decided to study direct crosses between A. cepa and A. fistulosum, knowing the problems with sterility that may occur in F1 progeny plants (and also of chromosome doubling by papers from other authors) and knowing of an alternative solution.

The problem of F1 and BC1 sterility of crosses between A. cepa  and A. fistulosum has remained since 1935.  As this reviewer rightly pointed out that an alternative method was proposed using an intermediate species A. roylei  between A. cepa and A. fistulosum to transfer desirable genes from A. fistulosum to A. cepa genome.  However, breeders from the World Vegetable Center had its own concept of creating bulb onion resistant to Stemphylium vesicarium. As we wrote in Introduction this work was initiated in 2001. Cytogenetic analysis were initiated in 2015, when World Vegetable Center asked our group to perform GISH and cytoplasmic analysis of advanced generation of hybrids between A. cepa and A. fistulosum showing resistance to Stemphylium vesicarium. Of course, it would be much more efficient with GISH to accompany the selection process at the very beginning.  But from this work there is a much more interesting phenomenon - spontaneous polyploidization and the absence of saturation with A. cepa genetic material  when backcrossing. The spontaneous chromosome doubling avoided the sterility problem, but created another problem, that is how to produce hybrids that would possess target gene from A. fistulosum in the complete A. cepa background.

Line 54: only across for chromosomal regions. Please leave out the word ‘for’

We made the correction.

Material and Methods

Line 68: An overview of the parental species, their hybrid (F5) that was used for backcrossing, and advanced generations of interspecific hybrids are given in Table 1.  Remark: This is confusing as according to the Table also F3 and F4 hybrid plants were used for backcrossing? Please adapt.

We analyzed all the original records of crosses and corrected the table.

Table 1. suggests the presence of plants in BC1F5 generation, BC2F5 generation and also BC2F4 and BC1F3 generations. Based on the GISH results it is clear that such plants did clearly not exist, as the chromosome number should be lower (eg 24) in case F5 plants were back crossed with onion plants. Since this is not the case in all but two genotypes, it is more likely that these plants occurred after apomixis or were selfings from AVON1275.

The plants existed but you are right according to the GISH results there was no evidence of backcrossing. In discussion we suggested possible reasons for the lack of backcrossing.

Line 78: Others hybrid lines did not flowering under our greenhouse conditions. Change into ‘flower’

Correction made.

Line 98: as those previously described, Leave out ‘those’

Correction made.

Line 108   We were able with GISH clearly distinguished the parental genomes. Change into We were able with GISH to clearly distinguish

Correction made.

Linea 109-112: ...despite the presence of highly identical DNA sequences in their genomes. It is known that A. cepa and A. fistulosum possessed subtelomeric tandem repeats with 80% of identity [20,21]. Also, retrotransposons are abundant in the Allium genomes [22], which are dispersed along entire chromosomes and may cause a probe in situ cross-hybridization. Please transfer these lines 109-112 to the discussion. This is not part of the results.

Correction made.

Line 113-115: In the hybridization mixture, we used a ratio of 1: 30 of probe and block DNA, and a 78% stringency washing was applied. GISH analyses were performed on 3-5 plants from each line of advanced interspecific hybrids. This information belongs to the M&M

Correction made.

Line 117: Leave out this part ‘which were used in backcrossing as a female parent,’ as it is not informative at that place

Correction made.

It was not revealed recombinant chromosomes between A. cepa and A. fistulosum. Please adapt this sentence

We replaced by “There were no recombinant chromosomes between A. cepa and A. fistulosum.”

Line 247: This study demonstrated that GISH is a powerful tool for identification of parental genomes at the chromosomal level in Allium interspecific hybrids. This has been shown before by Khrustaleva & Kik, not new.

Sentence removed.

Line 248: The spontaneous diploidization and absence of introgression into A. cepa during back-crossing were revealed in BC1F3, BC1F5, BC2F4 accessions. There is no absence of introgression, but results show that there is no evidence of back crossing! It is highly likely that in most plants apomixis has occurred (or selfing and then indeed without recombination).

We agree and made the correction.

Line 249: We suggested that diploidization caused by 2n-gamete formation due to second division restitution in meiosis. Explain: How would this be a matter of 2n-gamete formation? Both in the mother and in the father, meaning that both the egg cells and the pollen cells have a double number of chromosomes? Could this polyploidization not have happened after formation of the F1-zygote?

In the Discussion, we suggest possible mechanisms of polyploidization.

Line 250: Cytoplasmic determination showed the presence of bulb onion N cytoplasm in all analyzed hybrids. Our evaluation of both the nuclear and cytoplasm inherited genetic material makes very promising the use of the analyzed accessions in onion breeding.  Explain: Why is this of importance? Why is TA207 not included in the HRM analysis? The first cross was between TA207 and AF468, which finally resulted in AVON1275. So AVON1275 is expected to have the same cytoplasm as TA207. So, that all ‘back cross’ plants have the same cytoplasm as  progeny using AVON1275 is fully as expected. How does this result contribute to the paper or should it be discarded?

We believe that it is important to test the type of cytoplasm in the advanced generation of hybrids, despite the fact that according to the pedigree it should be known. However it was not clear if the bulb onion cytoplasm was normal (N) male-fertile or male-sterile (S) cytoplasm, which would be important for breeding of any future populations  We joined the work only in 2015. We were not involved in monitoring the process of crossing and selection from the very beginning. Therefore, it became necessary to check the type of cytoplasm.

TA207 was not included because we did not have DNA of this line

Add also information on how this result can be of interest to onion breeding and what needs to be done in the first place (haploidization).

We tried to make it clearer that cytogenetic evaluations should occur each generation to be sure that actual backcrossing has occurred, instead of carrying on self pollinations. 

Figures are not always clear explained: why are some genotypes stained with Cy3 and others with FITC? This should be described in the M&M. It is also not clear why in some genotypes especially the telomeres are more stained than the rest of the chromosomes. In some genotypes it almost looks as if only the telomers are stained green and not the rest of the chromosomes. More explanation is desired.   

In the beginning, we used the Biotin labeled probe and detected with Cy3, but then we switched to Digoxigenin labeled probe and detected with FITC because we got clearer results. We replaced the Cy3 picture with the FITC picture. Concerning pronounced signal at the distal end of the A. fistulosum chromosomes: It is known that most members of the Allium genus have a 375 bp common subtelomeric repeat. However, based on the GISH result we assume that A. fistulosum has, in addition to common subtelomeric repeat, its own specific subtelomeric repeat, which gives such a strong signal.

In the Discussion, attention should also be given to the fact that in AVON1291 the recombinant chromosome is in a homozyogus state (when did this occur?). 

We added statements explaining how this could happen to Discussion: In  AVON1291  (2n=4x=32; 16C + 15F + 1F/C) was revealed only one recombinant chromosome 2 of A. fistulosum possessing A. cepa segments at distal end , and its homolog  is  non-recombinant chromosome of A. fistulosum. It can be assumed that crossing over event between A. cepa and A. fistulosum chromosomes occurred at allotetraploid stage.  Genetic changes owing to homeologous recombination have been detected in both recently formed natural polyploids and synthetic interspecific polyploids (Gaeta & Pires  2009). Intergenomic pairing and recombination  have been documented for  allotetraploid  hybrids, for instance for sour cherry , Prunus cerasus,  for colchine-induced allotetraploid  hybrid between Lycopersicon esculentum x Solanum lycopersicoides , for  allopolyploids in the asterid clade (Crusz et al. 2017)

Round 2

Reviewer 1 Report

Critical comments:

Conclusion seems to be based on premature discussion. Perhaps, the most important output of this research was for this group to try to determine the chromosomal location of genes related to Stemphylium leaf blight (SLB) resistance gene(s) in Allium fistulosum. Due to the utilization of insufficient plant materials, it can only be concluded that the SLB gene(s) is located on A. fistulsoum chromosome(s) other than 5F.

(Major points)

1. M&M for analysis of (pollen) fertility and Table3

In order to show the credibility and reliability of pollen fertility data, the authors have to utilize an appropriate biological replicates. For instance, how many florets did the author apply to obtain the numerical data of pollen fertility?

 2. Statistics in Table2

Did the author carry out one-way ANOVA followed by Turkey-Kramer Test in order to show numerical data? What about biological replicates (n=4)?

(Minor points)

P2L50, P3L99: I(i)n situ -> should be Italic

P5L150, L163: P -> should be Italic

P7L189: missing2n=4x-2=30(16C+14F). -> missing (2n=4x-2=30) (16C+14F).

P12L325: Figure 3b, b’, b’’ -> Figures 3b, b’, b’’

Author Response

Critical comments:

Conclusion seems to be based on premature discussion. Perhaps, the most important output of this research was for this group to try to determine the chromosomal location of genes related to Stemphylium leaf blight (SLB) resistance gene(s) in Allium fistulosum. Due to the utilization of insufficient plant materials, it can only be concluded that the SLB gene(s) is located on A. fistulsoum chromosome(s) other than 5F.

The primarily goal of our research was not to map resistance to SLB.  We studied putative backcross progenies to determine if recombination had occurred among chromosomes of A. cepa and A. fistulosum, towards the long term goal of introgression SLB resistance.  GISH clearly supported spontaneous polyploidization and no evidence of backcrossing from interspecific hybrids between A. cepa and A. fistulosum. Considering that the attempts to transfer genes from A. fistulosum into A. cepa have not been successful since 1935 until the present time, for us it was a great interest to analyze genetic structure at chromosome level of advanced generation of interspecific hybrids between A. cepa and A. fistulosum showing high fertility and resistance to Stemphylium leaf blight (SLB). We did not attempt to map of genes related to SLB, for this we would need a combination of molecular markers and GISH. The inheritance of SLB resistance is not known; however Pathak et al. (2001) suggested that possibly a dominant gene conditions resistance. Missing of both homologous chromosomes 5 from A. fistulosum in the resistant hybrids is a strong indication that resistance to SLB is not associated with chromosome 5. 

(Major points)

1. M&M for analysis of (pollen) fertility and Table3

In order to show the credibility and reliability of pollen fertility data, the authors have to utilize an appropriate biological replicates. For instance, how many florets did the author apply to obtain the numerical data of pollen fertility?

We analyzed five flowers per accession when flowers were at peak anther maturity.  Two anthers from each flower were squashed in 1% acetocarmine on separate microscope slides and analyzed using phase-contrast microscopy with ocular x12 and objective x10 magnification.

 2. Statistics in Table2

Did the author carry out one-way ANOVA followed by Turkey-Kramer Test in order to show numerical data? What about biological replicates (n=4)?

One-way ANOVA was calculated based on four replications of each accession, and least significant differences were calculated using RStudio. 

(Minor points)

All corrections were made in the revision:

P2L50, P3L99: I(i)n situ -> should be Italic

P5L150, L163: P -> should be Italic

P7L189: missing2n=4x-2=30(16C+14F). -> missing (2n=4x-2=30) (16C+14F).

P12L325: Figure 3b, b’, b’’ -> Figures 3b, b’, b’’

Reviewer 3 Report

The revised version is acceptable for publication.

Author Response

 English language and style are fine/minor spell check required

We carefully checked the text and made corrections

Thank you for the positive evaluation of our work.